# Immune Checkpoints in Pediatric Solid Tumors: Targetable Pathways for Advanced Therapeutic Purposes

**DOI:** 10.3390/cells10040927

**Published:** 2021-04-17

**Authors:** Claudia Cocco, Fabio Morandi, Irma Airoldi

**Affiliations:** Laboratorio Cellule Staminali Post-Natali e Terapie Cellulari, IRCCS Istituto Giannina Gaslini, Via G. Gaslini 5, 16147 Genova, Italy; claudiacocco@gaslini.org (C.C.); fabiomorandi@gaslini.org (F.M.)

**Keywords:** immune checkpoint inhibitors, pediatric solid tumor, immune suppression

## Abstract

The tumor microenvironment (TME) represents a complex network between tumor cells and a variety of components including immune, stromal and vascular endothelial cells as well as the extracellular matrix. A wide panel of signals and interactions here take place, resulting in a bi-directional modulation of cellular functions. Many stimuli, on one hand, induce tumor growth and the spread of metastatic cells and, on the other hand, contribute to the establishment of an immunosuppressive environment. The latter feature is achieved by soothing immune effector cells, mainly cytotoxic T lymphocytes and B and NK cells, and/or through expansion of regulatory cell populations, including regulatory T and B cells, tumor-associated macrophages and myeloid-derived suppressor cells. In this context, immune checkpoints (IC) are key players in the control of T cell activation and anti-cancer activities, leading to the inhibition of tumor cell lysis and of pro-inflammatory cytokine production. Thus, these pathways represent promising targets for the development of effective and innovative therapies both in adults and children. Here, we address the role of different cell populations homing the TME and of well-known and recently characterized IC in the context of pediatric solid tumors. We also discuss preclinical and clinical data available using IC inhibitors alone, in combination with each other or administered with standard therapies.

## 1. Introduction

Tumorigenesis is a dynamic and complex process with characteristics that are responsible for tumor growth and dissemination. These peculiar features account for tumor complexity and consist of a wide variety of signals derived from different sources that all together promote uncontrolled cell division, angiogenesis, invasion and metastasis, resistance to apoptosis and evasion from immune surveillance. Different cellular and non-cellular elements within tumors, defining the tumor microenvironment (TME), are involved in all these processes.

The TME consists of non-malignant cells, including cancer-associated fibroblasts (CAF), endothelial cells, pericytes, immune and inflammatory cells, bone marrow-derived cells and components of the extracellular matrix (ECM) that establish a complex cross-talk with the tumor. The ECM is composed of collagen, proteoglycans and other molecules, including cytokines, growth factors (GF), hormones and chemical parameters (e.g., pH and interstitial pressure) regulating cancer progression. Furthermore, neoplastic cells have the ability to recruit and activate stromal cells, which in turn allow cancer cells to invade surrounding normal tissue and to metastasize in distant organs. Stromal cells also contribute to the formation and remodeling of ECM, produce several tumor growth factors and promote vessel formation [1,2]. 

The immune components within the TME are involved in both adaptive and innate immunity and are located in the core of the tumor as well as in the invasive margin or in the adjacent tertiary lymphoid structures. Basically all immune cell types may be found in the TME, including mast cells, neutrophils, macrophages (M1 and M2 polarized), myeloid-derived suppressor cells (MDSC), dendritic cells (DC), natural killer (NK), NKT cells and B and T lymphocytes. B cells include naïve and memory subsets, whereas T lymphocytes are predominantly represented by effector T helper (Th) cells including Th1, Th2 and Th17 cells, regulatory T (Treg) cells and follicular helper cells. All these immune cell populations have the ability to release a wide variety of cytokines, cathepsins, GF such as vascular endothelial GF (VEGF)-A and –C, fibroblast GF and epithelial GF, heparinases and matrix metalloproteinases(MMPs) that degrade ECM. All together these molecules promote cancer cell growth, metastasis and tumor vascularization. The cytokines released mainly promote an immune-suppressive microenvironment where IL-10 and TGF-β1 play a crucial role. TGF-β1, in particular, is produced by different cell populations, including tumor cells, endothelial and stem cells and fibroblasts, and (i) supports the growth and activities of CAF, (ii) stimulates angiogenesis and (iii) inhibits the functions of granulocytes, lymphocytes and antigen-presenting cells [3]. In addition, both IL-10 and TGF-β1 display immune-modulatory activities through different mechanisms including (i) activation of Treg cells recruited into the tumor, (ii) induction of a shift in the Th1–Th2 balance towards Th2 phenotypes without cytotoxic function, (iii) inhibition of Th1 responses, (iv) decrease in M1 activities paralleled by the stimulation of M2 functions and (v) induction of chemokine production (e.g., macrophage chemo-attractant protein 1) [4,5]. 

The importance of the crosstalk between the different cell populations within the TME and how it can impact on cancer progression has been clearly established, and represents an hallmark of cancer. The infiltration rate of different immune cells in the tumor correlate with cancer progression or patient prognosis. In this view, it is not surprising that an increased infiltration of cells with immune-suppressive activities such as Treg, MDSC and tumor-associated macrophages (TAMs) is associated with cancer progression [6], whereas the presence of cytotoxic T lymphocytes (CTL) correlates with a better prognosis in several cancers [7]. 

Due to the aforementioned considerations, TME cells have become a field of active investigation to develop novel therapeutic approaches, especially for those tumors unresponsive to first-line therapeutic protocols (chemotherapy/radiotherapy). 

## 2. Therapeutic Strategies Based on TME Immune Cells: A General Point of View

Different strategies have been developed over time to implement immunotherapeutic approaches against cancer, taking advantage of the population’s homing in the TME. Immunotherapy is focused on two primary aims, which are to strengthen the anti-tumor responses and to smother the immune suppressor mechanisms. These immunological activities may be obtained by using (i) monoclonal antibodies targeting tumor antigens, immune populations with suppressive or cytotoxic activities, (ii) immune-checkpoint (IC) inhibitors and (iii) adoptive cell therapy [8].

Monoclonal antibodies in cancer therapy are generally used in combination with chemotherapy and/or radiotherapy and function by three main mechanisms, which are (i)the targeting of molecules involved in tumor cell proliferation and angiogenesis, (ii) the induction of antibody-dependent cellular cytotoxicity (ADCC) and (iii)the activation of complement-dependent cytotoxicity (CDC). 

The development of IC inhibitors (ICI) is quite recent [9] and represents an important milestone in the field of immuno-oncology due to their ability to increase the power of immune responses against tumor growth [10,11]. This issue will be discussed in detail in the following chapters, but it is here to mention that impressive results have been achieved in several adult cancer patients [12,13,14,15]. These findings stimulated a prompt exploration of the ICI therapy also in childhood malignancies, in which data available are still limited.

Adoptive cell therapy actually represents the new frontier of immunotherapy and has undergone a continuous methodological evolution, starting from tumor-infiltrating lymphocytes (TIL), through TCR-editing, until the most revolutionary platform of chimeric antigenic receptors (CAR) T cells that account for a high number of clinical trials worldwide [16,17]. Here, we do not explore this complex and very wide topic since it is fully addressed in hundreds of reviews. Some of these approaches have shown promising results in adult tumors, paving the way for their use against pediatric malignancies, but, although childhood and adult cancers share some similarities, pediatric cancers show distinct features that may render therapeutic protocols commonly used in adult patients often inapplicable. In this context, it is important to mention that the TME of pediatric tumors displays higher immune-suppressive features than that of adult cancers, mainly due to the more abundant presence of TAMs, Tregs and MDSCs, especially in therapeutic-resistant pediatric solid tumors such as glioma, osteosarcoma and refractory neuroblastoma (NB) [18]. Compared to adult cancers, pediatric tumors also show fewer somatic coding mutations, resulting in a limited generation of neo-antigens, thus impairing T cell responses [19]. By contrast, similarly to adult cancers, the architecture of TME in childhood is characterized by disregulated vasculature and metabolic activity, that impair the trafficking and recruitment of anti-tumor effector cells [18]. 

Nonetheless, some immunotherapies used in adults received US Food and Drug Administration approval also for children. Here, we focus on immunotherapeutic approaches based on the use of ICI in pediatric tumors.

## 3. Immune Cells in the TME: A Double-Edged Sword

The cross-talk between cancer and immune cells is a three-phase process called “cancer immune-editing” [20]. First, before becoming clinically detectable, neoplastic cells are initially recognized and eliminated by reactive T cells which act against tumor-associated antigens (TAA). Afterwards, there is an “equilibrium phase” in which tumor cells remain under the control of the immune system, also for a long period, until more aggressive and less immunogenic neoplastic cells selectively survive and grow, entering a final phase characterized by T cell exhaustion and irreversible “immune escape” mechanisms. It is now well-known that cancer progression depends on the balance between promoting and antagonizing activities exerted by immune cells resident in the TME. Immunotherapy points to “shift the weight” on improving immune anti-tumor responses while impairing immune-suppressive mechanisms [20].

TME immune cell populations and, in some cases, their subsets, play different functions against harboring tumor cells that may be subverted during cancer progression. Cancer cells themselves may activate numerous immune escape mechanisms [21] which include (i)the loss of TAA expression, (ii)the down-regulation of the MHC complex on the cell surface, (iii)the production of molecules that inhibit antigen-presenting cell (APC) maturation (e.g., CCL2, CXCL1, CXCL5 and VEGF), (iv)the secretion of factors recruiting immune-suppressive cell populations (e.g., Treg, M2 macrophages and MDSCs), (v)the up-regulation of inhibitory receptors, including cytotoxic T lymphocyte-associated protein (CTLA)-4 and programmed cell death receptor 1 (PD-1), on T cells and (vi)the up-regulation of inhibitory ligands (PD-L1) on tumor or stromal cells. Immune therapies subverting these suppressive mechanisms, such as cytokine therapy and immune checkpoint inhibitors, have been developed and show clinical efficacy and long-term protection against several cancers [21].

### 3.1. T Lymphocytes

Antigen-specific CD8^+^T cells may infiltrate the tumor and represent the major effector T lymphocytes able to kill cancer cells by granule exocytosis, Fas ligand (FasL)-mediated apoptosis and by secreting interferon (IFN)-γ and tumor necrosis factor (TNF)-α [22]. Many strategies are employed by tumor cells to protect themselves against antigen-specific CTL, including, for example, the induction of negative regulators to inhibit CTL responses and the recruitment or polarization of immune-suppressive cells. The negative regulation of CTL is based on feedback mechanisms of the immune system, required under physiological conditions, to shut down immune responses, avoiding damage to bystander healthy tissues, once the antigen is eliminated. The prototype of these mechanisms is represented by IC molecules or co-inhibitory receptors. IC include both inhibitory and stimulatory molecules [23]. Stimulatory molecules are up-regulated uponT cell stimulation, whereas those with inhibitory activities repress different signaling pathways resulting in reduced activities of T lymphocytes, including cytotoxicity, secretion of cytokines and proliferation. Of note, T cells located in the TME present an intrinsic up-regulation of many immunosuppressive IC molecules and co-inhibitory receptors, paralleled by low levels of co-stimulatory molecules, a phenotype correlating with functional exhaustion or anergy [24]. Furthermore, a high infiltration of activated CTL has been described to be associated with improved clinical outcome in cancer patients affected by several tumors, including breast, ovarian, melanoma malignancies and non-small cell lung cancer (NSCLC) [25,26,27,28].

With opposite function from CTL are Treg [29] that are physiologically involved in the maintenance of immune homeostasis and peripheral tolerance. Treg suppress immune responses through different mechanisms including (i) metabolic disruption, (ii) direct B, CTL and NK cytolysis mediated by granzyme-B secretion, (iii) the inhibition of maturation and/or function of APC and (iv) the secretion of inhibitory cytokines such as IL-10 and TGF-β1 [30]. All these activities represent unambiguously an advantage for tumor growth, since they prevent the cytoxicity against cancer mainly operated by CTL. For this reason, the infiltration level of Treg in solid tumor correlates with poor prognosis of patients affected by epatocarcinoma [31], colorectal cancer [32] and NSCLC [33]. 

A peculiar Treg subset, strictly associated with the tumor, is the T regulatory type 1 (Tr1) cells. They are highly suppressive, produce IL-10 and TGF-β, and their generation from naïve CD4^+^ T cell precursors is promoted at tumor sites mainly through the activity of immature DC or tolerogenic plasmacytoid DC. This population may represent up to 30% of TIL and exert a suppressive activity up to 50 times more potent than Treg. Although Tr1 cells represent a highly suppressive Treg subset, their clinical impact in cancer is still uncertain [34,35]. 

A small fraction of T cells infiltrating TME is represented by γδ T cells, which exert anti-tumor activity mediated by the production of pro-inflammatory cytokines, direct cytotoxic activity and the regulation of the biological functions of other immune cell types. γδ T cells represent an important component of TIL in cancer patients [36,37] and are also considered as a good prognostic marker in many cancer studies [38]. 

### 3.2. NK Cells

NK cells are involved in innate immunity and exert powerful anti-tumor and anti-viral responses, by (i) the direct killing of tumor or infected cells through perforin and granzyme release, (ii) the stimulation of anti-tumor responses in other effector cells secreting pro-inflammatory cytokines such as IFN-γ, TNF-α, IL-6 and GM-CSF and (iii) recruiting DC and T lymphocytes through the production of chemo-attractant molecules (e.g., CCL5) [39,40]. Several studies reported that, in a variety of different solid tumors (lung, gastric, colorectal, head–neck and renal cell carcinoma), the presence of NK cells in the TME correlated with improved patient outcome [41]. By contrast, cells within the TME produce immunosuppressive cytokines and mediators that negatively affect NK cell functions. Immunosuppressive cytokines, such as TGFβ, IL-10 and IL-6, directly or indirectly inhibit NK cells. Of note, attenuated NK cell functions at the tumor site may be restored by stimulatory cytokines including IL-2, IL-15, IL-21 and IFN-α [42].

### 3.3. Dendritic Cells

DC are a rare heterogeneous population of leukocytes that play a crucial role in the induction and regulation of innate and adaptive immunity. DC are APC needed for the priming of effective T cell responses, their recruitment into the TME and the maintenance of effector memory T cell functions [43,44,45]. The activation of DC is mediated by different types of receptors expressed on their surface, named pattern recognition receptors (PRRs), including Toll-like receptors (TLRs), that recognize pathogen- or damage-associated molecular patterns such as nucleic acids released by dying tumor cells. DC capture TAA and process them into immunogenic peptides that are loaded onto MHC class I molecules and presented to CD8^+^ T cells. Thus, CD8^+^ CTL detect non-self tumor antigens expressed on cancer cells and, as already mentioned, kill them by producing cytotoxic cytokines, promoting granule exocytosis and FasL-mediated apoptosis. DC activation induces the up-regulation of co-stimulatory molecules (CD80, CD83 and CD86) and the production of inflammatory cytokines (type I IFNs and IL-12) that, in turn, mediate T cell priming and their differentiation into TAA-specific effector cells [46,47,48]. Moreover, DC induce the recruitment of effector T cells into the tumor site by secreting chemo-attractant molecules that contribute to maintain effector and memory T cell functions. By contrast, DC that are not activated promote immune tolerance [49,50].

### 3.4. B Lymphocytes

B lymphocytes contribute to the positive regulation of many processes associated with anti-tumor immunity [51]. Indeed, they express co-stimulatory molecules (e.g., CD40, CD80 and CD86), produce antibodies and cytokines, function as APC, initiate T-cell priming, promote T-cell expansion and memory differentiation [52,53,54] and directly kill tumor cells through FasL–Fas interaction [55] and the secretion of cytotoxic granules (e.g., granzyme B) [51]. Tumor-infiltrating B lymphocytes positively correlate with favorable clinical outcomes in different murine and human cancers [56,57,58]. Similarly to T lymphocytes, there are B cell subsets that exert immune-suppressive functions, namely regulatory B lymphocytes (Breg) [59,60] and tumor-associated B cells (TAB) [61].

The existence of Breg is physiologically related to the immune tolerance crucial to restrain the development of autoimmunity [62], but in TME they are expanded and support tumor growth by secreting IL-10 [63], IL-35 [64] and TGF-β that impair T cell proliferation and induce Treg [65]. Notably, a high frequency of Bregs correlates with the clinical stage of disease in hepatocellular carcinoma [66] and ovarian cancer [67] patients. The immunosuppressive role of TAB resembles that of Breg, since they promote tumor inflammation [68,69], inhibit anti-tumor T cell-dependent therapy responses [70,71] and produce IL-10 and TGF-β in mouse cancer models.

### 3.5. Macrophages

In physiological conditions, macrophages are polarized into two populations; M1 macrophages which are classically activated by interferon IFN-γ with lipopolysaccharides (LPS) and M2 macrophages, alternatively activated by IL-4 [72]. Activated M1 macrophages can kill tumor cells and cause tumor hemorrhagic necrosis [73] by producing reactive oxygen/nitrogen species. Their presence in the TME is associated with a favorable outcome in NSCLC [74], colorectal, prostate and other cancers [75,76].The activation of M1 macrophages may also occur in the early stages of tumorigenesis, when immune cells try to eliminate the nascent tumor [77]. By contrast, during tumor progression there is a subversion of macrophage functions, from M1 to M2, due to many factors, including the presence of IL-4 synthesized by CD4^+^ T cells and tumor cells [78,79] and of tumor-derived GF such as CSF1 [80] and GM-CSF [81]. M2 macrophages produce anti-inflammatory cytokines, promote angiogenesis and extracellular matrix degradation and are considered the major source of MDSC. MDSC exert different pro-tumor activities that include the induction of angiogenesis through (i) the production of MMP-9, prokineticin 2 and VEGF, (ii) the promotion of metastasis by producing arginase or inducible nitric oxide synthase (iNOS) and (iii) the inhibition of T cell functions through immunosuppressive cytokines, typically TGF-β and IL-10 [82,83].

Circulating monocytes give rise to mature macrophages, which are recruited into the TME and converted into tumor-associated macrophages (TAMs). TAMs represent up to 50% of normal cells in the TME and their phenotype is plastic and regulated by the local microenvironment. They are associated with tumor progression and have several properties similar to M2. TAMs secrete chemokines and cytokines (e.g., IL-6, IL-8 and IL-10) involved in the promotion of tumors and express several IC such as PD-1, CD47 and leukocyte immunoglobulin-like receptor 1 (LILRB1), and have been deeply investigated as a target for immunotherapeutic purposes [84].

High levels of PD-1 are expressed on TAMs and the level of PD-1 gradually increases with the development of tumors [85]. The PD-1/PD-L1 pathway is considered a tumor escape mechanism since it limits the activities of effector T cells, NK cells and DC, and inhibits the phagocytosis of TAMs. After PD-1/PD-L1 suppression by inhibitors, the phagocytosis of TAMs improves, thus killing tumor cells. Other immune checkpoints expressed on TAMs are signal regulatory protein α (SIRPα) and LILRB1, that bind CD47 and MHC class I, respectively, expressed on tumor cells. These engagements inhibit the phagocytosis of macrophages, thus promoting the occurrence and development of tumors. As a consequence of the use of drugs targeting such molecules, including anti-CD47, anti-SIRPα and anti-LILRB1 monoclonal antibodies, the recognition pathways are blocked and phagocytosis of macrophages is enhanced [86].

## 4. IC as Immunotherapeutic Targets in Pediatric Solid Cancers

The activation of immune effector cells in the TME is regulated by several activating and inhibitory molecules in a tightly coordinated network. The balance between these signals is crucial to activate an effective response against infected and neoplastic cells while maintaining immune tolerance against self-antigens [87]. 

IC include different trans-membrane proteins expressed by immune effector cells, mainly T lymphocytes, which regulate the intensity and duration of physiological immune responses, maintaining normal homeostasis and self-tolerance [88]. The expression of IC with immune-suppressive functions is increased on immune cells infiltrating the TME [23,89]. In addition, regulatory cell populations, such as MDSC, Treg, TAM and TAB, express inhibitory ligands of IC, thus leading to immune evasion mechanisms driven by cancer cells [23]. 

As already mentioned, during cancer progression a “cancer immune-editing”, based on three different phases, elimination, equilibrium and escape, takes place [20]. In the last phase, T lymphocytes become exhausted or tolerized, due to chronic antigen stimulation and the up-regulation of different inhibitory receptors [90]. The most characterized IC are the co-inhibitory molecules CTLA-4, PD-1 and its ligands PD-L1/2 [91], lymphocyte activating antigen-3 (LAG-3) [92], T-cell immunoglobulin and mucin-domain containing-3 (TIM-3) [93] and T cell immune-receptor with Ig and ITIM domains (TIGIT) [94]. Although not fully characterized, novel IC molecules have been discovered, such as B- and T-lymphocyte attenuator (BTLA) [95], B7-H3 [96] and indolamine dioxygenase (IDO) [97] that are under investigation for their potential clinical use in cancer patients. 

Pediatric solid tumors are a group of non-hematologic, extracranial cancers that occur during childhood. This heterogeneous group of tumors represents approximately 40% of all pediatric cancers. Many pediatric solid tumors are referred to as embryonal or developmental cancers because they arise in young children or adolescents as a result of alterations in the processes of organogenesis or normal growth. The ranking of pediatric cancer types is typical of western countries; leukemias are the most frequent neoplasms (33% of all malignant cancers), followed by lymphomas (16%), malignant tumors of the central nervous system (13%), neoplasms of the peripheral nervous system (8%) and of the soft tissues (7%). The remaining tumors are grouped primarily by anatomic site of onset and account for no more than 5% each and 23% altogether (AIRTUM consortium). Here, we mainly discuss IC pathways and results from pre-clinical and clinical studies obtained using IC inhibitors in pediatric cancers including medullo- and glioblastoma (derived from the central nervous system), and soft tissues such as rabdomiosarcoma, Ewing’s sarcoma (EWS) and NB, which represents the most common extra-cranial tumor in children. 

### 4.1. The PD-1/PD-L Axis

PD-1, also known as CD279, is a type I membrane protein belonging to the CD28 super-family and represents a key regulator of normal host physiology and of the programmed cell death of lymphocytes [98]. This is expressed upon the activation on different T cells subsets, B lymphocytes, NK cells, some myeloid cells and cancer cells [99]. The critical role of PD-1 in maintaining peripheral tolerance has been unambiguously demonstrated in PD-1 deficient mice which spontaneously develop autoimmune diseases, such as lupus-like proliferative arthritis and glomerulonephritis [100]. Ligands of PD-1 are two members of the B7 family, PD-L1 (CD274) and PD-L2 (CD273). PD-L1 has been found in a wide variety of cells located in primary and secondary lymphoid organs and in non-hematopoietic tissues, whereas PD-L2 is restricted to APC in lymphoid tissues. PD-1/PD-L1 interaction limits T cell activating signals by inhibiting T cell proliferation, survival and cytokine release [101].

The PD-1/PD-L1 axis has been deeply investigated in pediatric solid tumors. A high expression of PD-L1 was detected in alveolar rhabdomyosarcoma (86%), high-risk NB (72%), EWS (57%), embryonal rhabdomyosarcoma (50%) and osteosarcoma (47%). As expected, tumors with the highest proportion of PD-L1 positivity showed the poorest survival. In addition, CD8^+^ TIL significantly correlated with PD-1 expression and increased CD8^+^ TIL correlated with better overall survival, suggesting that triggering CD8^+^ T cell responses through PD-1/PD-L1 blockade would be a successful treatment strategy [102]. Majzner et al. [103] went on with this issue and analyzed the expression of PD-L1 on both tumor cells and tumor-associated immune cells (TAIC). They reported that the highest frequency (36%) of PD-L1 was observed in glioblastoma multiforme (GBM) and NB (14%) among all pediatric solid tumors analyzed. TAIC were represented by macrophages (20% of samples) and lymphocytes (72%), with a significant prevalence of macrophage infiltration in PD-L1-positive (51%) compared to PD-L1-negative tumors (17%). Notably, this macrophage population tested positive for PD-L1. This finding is consistent with the high rate of macrophage infiltration detected in pediatric solid tumors compared to adult [104,105]. Overall, PD-L1 was expressed on tumor cells and/or immune cells in 20% of samples. Moreover, the authors reported for the first time that PD-L1 expression on tumor cells and TAIC correlated with a worse overall survival of patients [103]. By contrast, other studies detected a low expression of PD-1, PD-L1 and PD-L2 in pediatric solid tumors [106,107]. 

PD-L1 was also found to be expressed on NB cell lines and primary metastatic neuroblasts isolated from bone marrow aspirates of high-risk NB patients with different MYCN amplification status. In addition, the presence of PD-1 on immune cells, including αβ and γδ T lymphocytes as well as NK cells, was clearly established [108]. Importantly, INF-γ stimulation de novo induced or up-regulated the expression of PD-L1 in freshly isolated metastatic neuroblasts from patients. Such induction showed a more rapid kinetics, compared to HLA-I molecules, thus suggesting that PD-L1 could limit the activity of T lymphocytes in advance before the acquisition of the HLA-I optimal level required for the KIR-mediated inhibition of NK cell functions [108].

Although the presence of TIL had a positive clinical impact in high-risk NB patients [109], Melaiu and coworkers [110] correlated a worse prognosis in NB patients with the presence of PD1^+^ and LAG3^+^ TIL and a high density of PD-L1^+^ and HLA class I^+^ tumor cells in the TME. The authors identified two PD-L1/HLA-I combinations, irrespective of T cell infiltration level, MYCN amplification status, stage of disease and age at diagnosis: the first, characterized by high HLA-I and low/negative PD-L1, was associated with a good prognosis, whereas the second, represented by low HLA-I and high/negative PD-L1, correlated with a poor prognosis. Thus, the combined analysis of PD-L1/HLA-I expression represents a predictive biomarker of clinical outcome for NB patients. Notably, the silencing of MYC and MYCN oncogenes led to a down-regulation of PD-L1 expression in NB cells, both in vitro and in vivo, thus suggesting that the pharmacological inhibition of this axis may be used as a therapeutic strategy in high-risk NB patients [109].

The expression of PD-1 and PD-L1, along with the prognostic relevance of TIL, was also investigated in pediatric gonadal germ cell tumors, a heterogeneous group of tumors which represent 3–5% of all childhood cancers occurring before 15 years of age, and 15% of neoplasms in adolescents aged 15–19 years [111]. Three different cancer phenotypes were identified that were tumors (i) with no T cell infiltration, (ii) highly infiltrated by PD-1^+^ CD8^+^ T cells and (iii) highly infiltrated by CD8^+^ T cells within an immunosuppressive TME characterized by Treg cells and PD-L1^+^ neoplastic cells. TIL influenced the progression of gonadal germ cell tumors and showed clinical relevance to improve the risk stratification and treatment of pediatric patients, whereas PD-L1 showed a different prognostic value when expressed on tumor cells or TIL [112]. Another study carried out by Chovaneck et al. [113] reported that patients with testicular germ cell tumors had the worst prognosis in the presence of PD-L1^hi^ tumor cells and PD-L1^low^ TILs. On the contrary, PD-L1^low^ tumor cells and PD-L1^hi^ TILs predicted a better prognosis. 

Expression of PD-L1 was further investigated in pediatric soft tissue sarcoma (STS). Kim et al. performed a tissue micro-array analysis of PD-L1 expression in neoplastic cells from rhabdomyosarcoma, synovial sarcoma, EWS, epithelioid sarcoma and mesenchymal chondrosarcoma. PD-L1 was expressed in 43% of these tissues, with a significant difference between histological subtypes of sarcoma. The proportion of PD-L1^+^ tumors was highest in epithelioid sarcoma (100%), followed by synovial sarcoma (53%), rhabdomyosarcoma (38%) and Ewing sarcoma (33%), whereas mesenchymal chondrosarcoma tested negative for PD-L1 [114]. More importantly, high PD-L1 expression was significantly associated with worse overall survival, regardless of sex, age, tumor size, histology, site, surgical outcome and adjuvant treatment, thus envisaging that, similarly to other pediatric tumors, the PD-1/PD-L1 axis may represent a potential therapeutic target for the treatment of young STS patients [114]. Another study reported the clinical impact of PD-1/PDL-1 expression and TIL infiltration in sarcoma sub-types. Van Erp et al. analyzed PD-L1 expression in biopsies from a wide panel of primary untreated osteosarcoma, EWS, alveolar rhabdomyosarcoma, embryonal rhabdomyosarcoma, synovial sarcoma and desmoplastic small round cell tumors (DSRCT). PD-L1 was predominantly detected in alveolar and embryonal rhabdomyosarcomas (15 and 16%, respectively) and was predictive of a better event-free and metastases-free survival in alveolar sarcomas. Furthermore, infiltration of PD-1^+^ lymphocytes was mainly observed in synovial sarcomas (18%), whereas a high infiltration of CD8^+^ lymphocytes was detected mostly in osteosarcomas (35%) and correlated with a worse event-free survival. EWS and DSRCT showed PD-1^+^ tumor cells but not PD-1^+^ TIL [115]. The latter observation is in accordance with results obtained by Spurny, that reported the PD-1/PD-L1 axis was not involved in EWS [116]. PD-L1 expression was significantly associated with an increased infiltration of T lymphocytes, DC and NK cells in osteosarcoma patients and correlated with a worse prognosis. In particular, infiltration by DC and macrophages was associated with a worse event-free survival at five years [117]. 

Different authors tested the expression of PD-L1 in pediatric tumors of the central nervous system (CNS). A study focused on glioma reported that PDL-1 was over-expressed on ki-67-negative tumor cells and such expression was significantly increased in high-grade as compared to low-grade gliomas. Moreover, the lack of PDL-1 expression on tumor cells correlated with higher TIL infiltration [118]. In another study, a high lymphocyte infiltration and the presence of infiltrating PD-1^+^ CD4^+^ and CD8^+^ T cells, represented a favorable prognostic marker in human papillomavirus-infected head and neck cancer and not in those uninfected [119]. Children affected by medulloblastoma displayed a limited number of PD-1^+^ T cells and low to absent levels of PD-L1, with the exception of the sonic hedgehog subtype [120]. In patients with pediatric ependymomas, PD-L1 expression was detected only in supratentorial tumors expressing RELA fusion protein, both in tumor and myeloid cells, whereas PD-1 expression was detected on both CD4^+^ and CD8^+^ infiltrating T lymphocytes. By contrast, other ependymoma subtypes showed low PD-L1 expression, with no prognostic significance [121]. 

#### Preclinical and Clinical Studies Targeting the PD-1/PD-L1 Axis in Pediatric Tumors

Most preclinical studies focusing on the blockade of PD-1/PD-L1 axis in pediatric solid tumors have been carried out in glioblastoma. Wainwright and coworkers [122] setup an orthotopic model based on intracranial injection of glioblastoma cell lines and treatment with blocking antibodies against PD-L1 or other immune checkpoints, such as CTLA-4 and IDO. Although prolonged survival was observed in mice targeting uniquely PDL-1, the best results were achieved by a simultaneous block of these three immune checkpoints. Similarly, it has been demonstrated that combined therapy using anti-PD-1 and anti-TIGIT [123] or anti-TIM-3 [124] antibodies significantly prolonged mice survival compared to untreated mice and mice treated with single therapies. In addition, it was reported that a significant increase in the therapeutic effect of anti-PD-L1 antibodies was obtained by associating an agonist of Toll-like receptor 3 (TLR3), that activated DC and increased infiltration of immune effector cells within the tumor, [125] or standard chemotherapy [126]. Another interesting pre-clinical approach [127] is represented by the use of gene-mediated cytotoxic immunotherapy (GMCI) that leads to the increase in PD-L1 expression in glioblastoma both in vitro and in vivo. This resulted in increased T lymphocyte infiltration in an orthotopic preclinical model of glioblastoma, although the survival of mice was significantly prolonged by combined therapy using GMCI and anti-PD-L1 antibodies. Collectively, these studies demonstrate that therapeutic strategies targeting PD-1/PD-L1 axis may be promising and effective for patients with glioblastoma, in particular when combined with standard therapies or with the blockade of other immune checkpoints. Several antibodies targeting PD-1 or PD-L1 have received clinical approval as first- and second-line treatments for different malignancies and numerous clinical trials are ongoing to test the efficacy of these drugs when used alone or in combination with conventional anti-cancer drugs, as well as targeted therapies [128]. Although clinicians may choose from several IC inhibitors to disrupt PD-1 in adult cancer patients, this therapeutic option is still limited in childhood. To date, five anti-PD-1/L1 antibodies have been approved by the FDA, which are nivolumab and pembrolizumab against PD-1 and avelumab, atezolizumab and durvalumab against PD-L1. Many others are under FDA approval. Phase I and, to a lesser extent, phase II clinical trials are in progress and are mainly recruiting pediatric patients with glioblastoma or different solid tumors that are treated with anti-PD1 drugs in combination with conventional therapies, radio-chemotherapies or surgery. Of note, a bi-functional fusion protein which targets simultaneously PD-1 and TGF-β has been successfully used in preclinical models, displaying an increased anti-tumor activity as compared to drugs which target PD-1 alone [129]. This molecule, named M7824, is currently utilized in several clinical trials for adult solid tumors and may represent a future therapeutic strategy also for pediatric malignancies.

Combined therapies are currently being investigated in 35 clinical trials involving glioblastoma patients (www.clinicaltrials.gov, accessed on 28 March 2021). By contrast, although no preclinical study has been carried out in NB, two clinical trials are currently ongoing using anti-PD-1 antibodies or PD-1 specific inhibitors, in combination with anti-GD2 antibodies or other drugs. 

Figure 1 summarizes the PD-1/PD-L1 axis and the activities of blocking antibodies.

### 4.2. CTLA-4

CTLA-4, also named CD152, is an inhibitory receptor belonging to the immunoglobulin (Ig) super-family, which negatively regulates T cell responses, thus avoiding the generation of potential auto-reactive T cells in the early activation phase [130]. CTLA-4 is structurally homologous to the co-stimulatory molecule CD28 and competes with the latter molecule for the same ligands. CTLA-4 binds to CD80 and CD86 with greater affinity and avidity than CD28, thus reducing the risk of aberrant T-cell activation and potential self-reactions [131]. Indeed, such interaction leads to the inhibition of IL-2, IFN-γ and IL-4 production and dampens the expression of the IL-2 receptor on T lymphocytes, thus leading to a decreased activation and proliferation of T cells and the induction of apoptosis [132]. Of note, CTLA-4 is virtually absent on the surface of naïve T cells, but is present on memory and effector CD4+ and CD8+ T cells and on Treg [133,134]. The role of CTLA-4 in the control of auto-reactive T cells has been demonstrated in CTLA-4 deficient mice, who developed a lethal form of lymphoproliferative disorder [135]. In addition, a soluble form of CTLA-4, generated through alternative mRNA splicing, has been reported to inhibit early T-cell activation [136]. Moreover, high serum levels of soluble CTLA-4 are associated with the onset of several autoimmune diseases, including systemic lupus erythematosus, rheumatoid arthritis, Grave’s disease, autoimmune hypothyroidism and type 1 diabetes [137]. 

CTLA-4 expression was detected in different adult cancers and in pediatric solid tumors, such as glioblastoma and NB [138]. A high rate of infiltrating Treg expressing CTLA-4, with increased suppressive functions, was detected in glioblastoma patients [139]. Moreover, children affected by aggressive sarcomas display a higher expression of CTLA-4 in circulating CD4 and CD8 T lymphocytes [140]. 

The effects of anti-CTLA-4 treatment were investigated in preclinical studies. In a subcutaneous model of human NB, an anti-CTLA-4 antibody was administered in combination with an anti-GD2 antibody and radiotherapy. In this setting, the combined therapy induced a significantly increased overall survival and reduction in tumor growth, as compared to mice treated with single therapies [141]. Similar effects were observed in an orthotopic model of glioblastoma, in which anti-CTLA-4 alone displayed only marginal effects in terms of reduction in tumor growth and prolonged overall survival. However, once administered a combination of anti-CTLA-4 and anti-PD-1 antibodies, therapeutic effects were improved due to the increased infiltration of effector T cells, paralleled by a decreased infiltration of Treg [142]. Similar results were obtained in an orthotopic model of glioblastoma where the highest curative effect was obtained using a combination of anti-CTLA-4 and anti-PD-1 antibodies in addition to oncolytic viruses. In this case, the anti-tumor activities were related to macrophage polarization and the increase in the ratio between infiltrating effector T cells and Treg [143]. These studies confirm that CTLA-4 blocking may revert tumor-mediated immunosuppression, thus rendering more effective other immunotherapeutic strategies. Recently, a bi-specific molecule which targets simultaneously CTLA-4 and TGF-β (a-CTLA4-TGFβRII) has been tested in preclinical models of adult solid tumors, showing an increased anti-tumor activity as compared to drugs targeting CTLA-4 alone, due to Treg inhibition [144]. This approach may represent a promising therapeutic strategy also in the context of pediatric solid tumors. 

To date, four clinical trials involving children affected by NB and nine in glioblastoma patients are currently ongoing using anti-CTLA-4 antibodies in combination with standard therapies or other drugs (www.clinicaltrials.gov, accessed on 28 March 2021).

The functional consequences of CTLA-4 triggering and of blocking the CTLA-4/CD80/CD86 axis are reported in Figure 2.

### 4.3. B7-H3

B7-H3 (CD276) is a glycoprotein encoded by the CD276 gene located on chromosome 15, which belongs to the B7 family of molecules, mainly expressed on APC and involved in the inhibition of T cells (Figure 3). Although the B7-H3 receptor remains an orphan ligand, a potential receptor on activated immune cells was represented by TLT-2 [145]. B7-H3 was initially described as a stimulator of T cell responses and IFN-γ production [146], but its role in immune evasion, through the inhibition of CD4+ and CD8+ T cell proliferation was next discovered [147], making B7-H3 an interesting target for new immunotherapeutic treatments.

B7-H3 plays a role in cancer progression not only by mediating immune evasion, but also by promoting migration, angiogenesis, gene regulation via epigenetic mechanisms [148,149,150] and enrichment of cancer stem cells [151]. Recent pre-clinical studies suggested that B7-H3 over-expression impacts drug resistance, since B7-H3 depletion enhanced the chemo-sensitivity of cancer cells to chemotherapeutic drugs in melanoma and breast cancer.

B7-H3 was identified in NB as a highly specific marker of tumor cells which inhibits NK cell-mediated lysis [152] and predicts a worse prognosis [153]. Recently, the largest screen of B7-H3 expression in pediatric tumors revealed a high and homogeneous expression in EWS, rabdomyosarcoma, NB, Wilms’ tumor and medulloblastoma [154]. In osteosarcoma patients, its high expression is inversely correlated within filtrating CTL in the TME and predicts worse prognosis [155], whereas in glioblastoma it mediates invasiveness and immunosuppression [156,157].

A recent preclinical study tested the efficacy of a drug-conjugated anti-B7-H3 antibody in patient-derived and cell line-derived xenografts of EWS, rhabdomyosarcoma, Wilms’ tumors, osteosarcoma and NB. Promising results were obtained in terms of overall response (91.5%) and complete response (64.4%), thus confirming that B7-H3 represents a useful therapeutic target for different pediatric solid tumors [158].

B7-H3 CAR T cells have been recently developed and tested in a wide panel of pediatric tumors with promising results in preclinical models, especially in terms of prolonged survival [154]. Similar results were also obtained using CAR T cells with double specificity for B7-H3 and GD2 [159] in preclinical models of NB.

Three clinical studies using B7-H3-specific CAR T cells in combination with temozolomide (TMZ) are currently recruiting patients with glioblastoma and other CNS tumors. In addition, three clinical studies are recruiting NB patients, two of them using B7-H3 specific CAR T cells and the other using anti-B7-H3 antibody enoblituzumab. Finally, three clinical trials are ongoing for patients with STS, one based on enoblituzumab, one using B7-H3 specific CAR T cells and the last using a bi-specific molecule which targets B7-H3 and CD3 (www.clinicaltrials.gov, accessed on 28 March 2021).

Figure 3 depicts the role of B7H3 in cancer immunosuppression and the effects of blocking antibodies and CAR T cells.

### 4.4. LAG-3

Lymphocyte activation gene (LAG)-3 is a checkpoint molecule composed of a trans-membrane protein belonging to the Ig super-family and four extracellular domains known as D1–D4. LAG-3 shows a structural homology with CD4 and interacts with the same ligand, the MHC class II molecule, but binds to the stable complex with a higher affinity than CD4. The interaction of LAG-3 with the MHC class II molecule inhibits CD4+ T cell proliferation and cytokine release (Figure 4) [160], whereas that with other ligands present in the TME, such as galectin-3, fibrinogen-like protein 1 and liver sinusoidal endothelial cell lectin, delivers regulatory signals in CD8+ T lymphocytes and NK cells [160]. LAG-3 is expressed on activated CD4+ and CD8+ T cells, Treg, NK and B cells, TIL and plasmacytoid DC [161,162,163,164,165]. As for other checkpoint molecules, the role of LAG-3 has been clearly established using KO mice [166]. 

The expression and function of LAG-3 in the TME was investigated in glioblastoma by Harris-Bookman and co-workers [167], who demonstrated its expression mainly in TIL. Moreover, they setup an orthotopic preclinical model of glioblastoma and observed that combined therapy using anti-LAG-3 and anti-PD-1 antibodies was more effective in terms of increased overall survival than the single therapy. The expression of LAG-3 was also analyzed in STS, where it was predominantly detected in TIL. In addition, high LAG-3 expression correlated with a worse prognosis of STS patients. 

To date, two clinical trials are ongoing using anti-LAG-3 antibodies as a therapeutic strategy for children affected by glioblastoma. Both studies adopted these antibodies in combination with the anti-PD-1 antibody (Nivolumab, www.clinicaltrials.gov, accessed on 28 March 2021)

### 4.5. TIM-3

TIM-3 is a type I trans-membrane protein, belonging to the Ig super-family, composed of an N-terminal IgV domain, a mucin-like domain with glycosylation sites and a C-terminal cytoplasmic domain with two out of five tyrosine residues, whose phosphorylation is related to TIM-3-mediated signalling [168]. In addition to its inhibitory function, TIM-3 seems to play a co-stimulatory function, highlighting its dual role in immune responses [168]. TIM-3 was first identified in Th1 and cytotoxic CD8+ T cells in 2002 [169] and afterwards several studies reported an association between TIM-3 gene polymorphisms and the risk of developing autoimmune diseases (i.e. Hashimoto’s disease, idiopathic thrombocytopenic purpura, multiple sclerosis and rheumatoid arthritis) [170]. 

The ligands of TIM-3 are galectin-9 (Gal-9), phospatidyl serine (PtdSer), high-mobility group protein B1 (HMGB1) and Carcino embryonic Antigen Related Cell Adhesion Molecule 1 (Ceacam-1). Upon ligation, these molecules induced apoptosis (Gal-9), exhaustion (Ceacam-1) or impairment of activation (HMGB1) in T lymphocytes (Figure 4). In contrast, the ligation of PtdSer, which is over-expressed in apoptotic cells, induced the clearing of apoptotic bodies and the reduction in antigens’ cross-presentation by DC [168]. 

Different studies addressed the presence of TIM-3 in pediatric tumors and underlined its role in the suppression of anti-tumor immune responses. Goods and coworkers [171] analyzed glioblastoma tissues and reported a clear infiltration of PD-1+TIM-3+ lymphocytes that expressed several markers of exhaustion, thus suggesting the therapeutic use of antibodies blocking these two immune checkpoints. Other studies demonstrated the presence of TIM-3 in glioma/glioblastoma TMEs and that such expression may be used as a prognostic factor of poor prognosis [172,173]. In this view, Zhang and coworkers revealed that a low expression of TIM-3, which is related to MGMT promoter methylation, was predictive of a better clinical outcome [174]. It has been also shown that TIM-3 is present in different STS and exerts immunosuppressive functions through different mechanisms, which are (i) the decrease in proliferation and secretion of pro-inflammatory cytokines by TIL, (ii) the induction of anergic T cells related to the presence of CD163+ M2 macrophages in TME, (iii) the up-regulation of epithelial–mesenchymal transition markers and (iv) the induction of the proliferation of tumor cells, thus suggesting that TIM-3-blocking antibodies may inhibit tumor growth [175,176,177,178]. Furthermore, the expression of TIM-3 in osteosarcoma tissue [179] and the presence of soluble TIM-3 in serum samples of patients [180] correlated with worse prognosis. Other studies reported that TIM-3 is expressed in peritoneal monocytes, macrophages and DC in patients affected by histiocytic sarcoma and other histiocytic and dendritic cell neoplasms [181,182].

Preclinical studies have been carried out using anti-TIM-3 blocking antibodies. In a preclinical model of glioma, the treatment with anti-TIM-3 and anti-Ceacam-1 significantly increased the overall survival of mice, but the best results were obtained combining the two antibodies (Figure 4). Such a therapeutic effect was paralleled by increased effector T cells infiltration within the tumor and the decreased presence of Treg. Finally, plasma levels of IFN-γ and TGF-β were increased, thus witnessing an activation of anti-tumor immune responses [183].

Blocking antibodies against TIM-3 are currently used in combination with anti-PD-1 in a phase I/II clinical trial for patients affected by glioblastoma multiforme (www.clinicaltrials.gov, accessed on 28 March 2021).

### 4.6. TIGIT

TIGIT is a cell surface protein identified by bioinformatics analysis of the genes expressed in activated T cells. TIGIT is composed of a single Ig domain, a type I trans-membrane domain and a single intracellular immune-receptor tyrosine-based inhibitory motif (ITIM). It belongs to the poliovirus receptor family and, together with the co-stimulatory molecules CD96 and CD226 (DNAM-1), forms a pathway similar to that of CTLA-4/CD28. TIGIT is expressed on the surface of αβ T cells upon activation, memory T lymphocytes, Treg and NK cells in which it inhibits NK cell-mediated killing [184]. TIGIT can interact with at least three ligands, namely CD155, CD112 and CD113, belonging to the family of nectin/NECL molecules (Figure 4). All these molecules mediate cell adhesion, cell polarization and tissue organization, and may also function as receptors for herpes- and poliovirus [185]. CD155 is mainly expressed on DC, T and B cells and macrophages (Figure 4) but also in non-hematopoietic tissues such as kidney, nervous system and intestines, whereas CD112 has a wide expression in both hematopoietic and non-hematopoietic tissues (i.e., bone marrow, kidney, pancreas and lung) and CD113 is restricted to non-hematopoietic tissues (i.e., placenta, testis, kidney, liver and lung). CD155 displays a higher affinity for TIGIT than CD112 or CD113 and may also interact with CD96 and DNAM-1 [185], thus impairing the co-stimulation of DNAM-1 and delivering inhibitory signals through the inhibition of ERK activation in DC and the impairment of T cell responses. Moreover, TIGIT regulates T cell functions by activating Treg (Figure 4) [185].

In cancer, TIGIT has been detected in glioblastoma TME, mostly localized in the tumor core [186] and TIL [123,187]. In osteosarcoma, TIGIT+ T cells are highly present and, when treated ex vivo with an anti-TIGIT antibody, acquire cytotoxic activity against neoplastic cells in vitro, thus supporting the potential clinical application of TIGIT blockade for children affected by this cancer [188].

Anti-TIGIT antibody, so far employed in preclinical studies only in combination with anti-PD-1 antibody, showed promising results in glioblastoma animal models since an increased survival was observed and correlated with the infiltration of effector T cells in the tumor [123] (Figure 4). Such combined therapy is currently under investigation in a phase I clinical trial for patients with glioblastoma (www.clinicaltrials.gov, accessed on 29 March 2021).

### 4.7. IDO-1

Indoleamine 2,3-dioxygenase (IDO)-1 is a tryptophan (Trp) catabolic enzyme which can be classified as an IC due to its immune-inhibitory properties (Figure 4) [189]. Trp is an essential amino acid for neuropsychological and immunological functions and is expressed not only by DC and MDSC, but also by tumor cells, endothelial cells, fibroblasts and immune cells infiltrating the TME (Figure 4) [190]. IDO-1 expression in the immune cells of the TME is regulated by different factors secreted by tumor cells (i.e., Wnt5 and sTGFBR3). On the other hand, pro-inflammatory cytokines released by immune effector cells, such as IFN-γ, IL-6, TNF-α, IL-1 and TGF-β, up-regulated IDO-1 on tumor cells, thus representing an immune escape mechanism [190].

IDO immune suppression takes place through the metabolic depletion of Trp and/or the accumulation of kynurenine (Kyn), leading to (i) the inhibition of T and NK cell effector functions and (ii) the activation and induction of Treg and MDSC (Figure 4) [191]. Such IDO-1-mediated modulation of innate immune responses was initially demonstrated in infectious diseases [192,193], but further evidence identified IDO as a mediator of immune tolerance [194]. In addition, IDO-1 is involved in cancer vascularization, metastasis [195] and cancer progression (Figure 4) [196], thus envisaging a target for potential immunotherapies. IDO expression in glioblastoma correlates with the progression and recruitment of Treg in the TME [197,198,199,200], whereas in osteosarcoma it correlates with poor prognosis [201].

Preclinical studies using glioma cells revealed that the treatment of mice with IDO inhibitors prolonged survival and limited tumor growth, with a higher effect when used in combination with TMZ. Superimposable results were obtained using tumor cells with knockdown of IDO [202], as well as using the IDO inhibitor PCC0208009 [203]. Moreover, the pharmacological inhibition of IDO increased the anti-tumor effects of radiotherapy [204]. Finally, IDO inhibitors increased the survival of tumor-carrying mice when combined with anti-PD-1/anti-CTLA-4 antibodies, through the reduction in Treg in the TME and the increase in anti-tumor effector cells [122].

Five clinical trials are currently ongoing for patients with glioblastoma using the pharmacological inhibition of IDO-1 in combination with TMZ and/or anti-PD-1 antibody (www.clinicaltrials.gov, accessed on 29 March 2021).

### 4.8. BTLA

BTLA is a surface molecule composed of two immune-receptor tyrosine-based inhibitory motifs in the cytoplasmic region, and represents an immune-suppressive checkpoint [205]. BTLA is expressed on T and B lymphocytes, macrophages and DC and NK cells, and regulates inflammatory responses by affecting TCR γδ T cell homeostasis, CTL activity and the production of inflammatory cytokines [206,207]. BTLA inhibitory activities were demonstrated in BTLA−/− lymphocytes that showed an increased proliferation [208].

Herpes virus entry mediator (HVEM), a receptor of the tumor necrosis factor family, is the known ligand for BTLA in mice and humans. The binding of BTLA to HVEM has been shown to recruit Src homology 2 domain-containing protein tyrosine phosphatase (SHP)-1 and -2 proteins, resulting in the suppression of T cell receptor (TCR) activation [205,209].

BTLA plays inhibitory roles in several experimental study models, including encephalomyelitis, colitis and major histocompatibility complex-mismatched cardiac allograft by modulating T cell responses [210,211]. In addition, BTLA may attenuate B cell functions and prevent NKT cell-mediated hepatitis [212]. Increased expression of BTLA and HVEM in gastric cancer were found to be associated with progression and poor prognosis [213,214]. 

So far, only one study characterized the expression of BTLA and its receptor HVEM in different pediatric solid tumors. This study demonstrated that HVEM is expressed in almost all rhabdomyosarcoma and osteosarcoma samples. Moreover, 45% of rhabdomyosarcoma and osteosarcoma samples expressed both HVEM on tumor cells and BTLA on TIL, thus suggesting that the BTLA/HVEM axis may be involved in immunosuppression in these pediatric tumors [215].

## 5. Conclusions

IC blockade is indisputably an emerging cancer treatment, but an important issue to be considered for the development of IC inhibitor-based therapy is the identification of predictive biomarkers for selecting patients which are potentially responsive to this therapeutic approach. IC expression profile and the infiltration rate of immune cells, in particular of TIL, have been suggested as potential prognostic factors, both in adult and childhood cancer patients. So far, the expression profile of IC in pediatric cancer patients is heterogenous among different tumors and in different subtypes of the same cancer, providing confusing results. Such variability can be ascribed to (i) non-standardized methodologies for IC detection and to (ii) the type of cells analyzed (i.e., tumor cells and/or infiltrating immune cell). Thus, these limitations provided an incomplete overview of tumors potentially targeted by ICI. To complicate this scenario, pediatric cancers show reduced production of neo-antigens. Consequently, TIL are lower than in adult tumors and/or are usually trapped by tumor stromal components, resulting in them being ineffective in the control of tumor growth [104]. For these reasons, therapeutic strategies aimed at recruiting TIL within inaccessible sites of the TME [216] may be more successful. Furthermore, the efficacy of ICI can be prejudiced by the presence of immune-suppressive cells in the TME. In this context, a full characterization of the TME cell components of each cancer type may help to identify specific prognostic and therapeutic biomarkers, and to design effective combined therapies. Indeed, the administration of ICI as a mono-therapy was partially efficacious, whereas their association with other ICI, chemo/radiotherapy, T cell-based therapy and monoclonal antibodies substantially improved the clinical outcome of adult patients and are currently being investigated in pediatric clinical trials [217].

IC blockade may have better safety compared to chemotherapy, but some immune-related adverse events occurred, thus requiring specialized management of patients. This is intrinsically due to the induction of inflammatory side effects, often named immune-related adverse events. Although any organ system may be affected, the immune-related adverse events primarily involve the skin, gastrointestinal tract, endocrine glands and liver and, more marginally, the hematologic, central nervous and cardiovascular systems. The wide range of potential immune-related adverse events requires multidisciplinary, collaborative management by providers across the clinical spectrum, especially in the setting of pediatric patients. However, it is important to mention that most of these adverse events are reversible, with the exception of side effects on the endocrine system, and deaths are exceptionally rare. However, to overcome this toxicity some questions need an answer, which are (i) why do immune-related adverse events occur? (ii) why do these toxicities appear in some patients and not others? (iii) are adverse events associated with the efficacy of the treatment? (iv) is it safe to restart immune checkpoint blockade after serious adverse events? (v) is it necessary to restart immune checkpoint blockade after event resolution? and (vi) is it safe to treat patients at increased risk for these treatments [218]? However, there are important opportunities to improve the treatment of immune-related adverse events that include a deep investigation of the mechanisms of immune-related adverse events (e.g., events mediated by antibodies, cytokines and T lymphocytes) and the sharing of data related to immune-related adverse events in patient populations that are underrepresented in clinical trials. In this way, it will be feasible to develop more precise treatments for immune-related adverse events and to realize the full potential of this treatment approach. Accordingly, the use of bi-specific molecules, or bi-specific antibodies, targeting IC and immunosuppressive cytokines, may represent a promising tool to increase anti-tumor activity and to limit adverse events. Indeed, nearly 200 clinical trials are currently investigating the role of these molecules in adults affected by solid tumors, thus suggesting a promising future application for pediatric tumors. In addition, a recent study used CRISPR-Cas9 technology to suppress PD-1 expression on CAR T cells, thus enhancing their anti-tumor activity [219]. This study may pave the way to the use of this technology on engineered T cells to disrupt the interaction between IC and their ligands, avoiding the use of drugs in combination with adoptive cell therapy, thus circumventing possible adverse effects.

## Figures and Tables

**Figure 1 cells-10-00927-f001:**
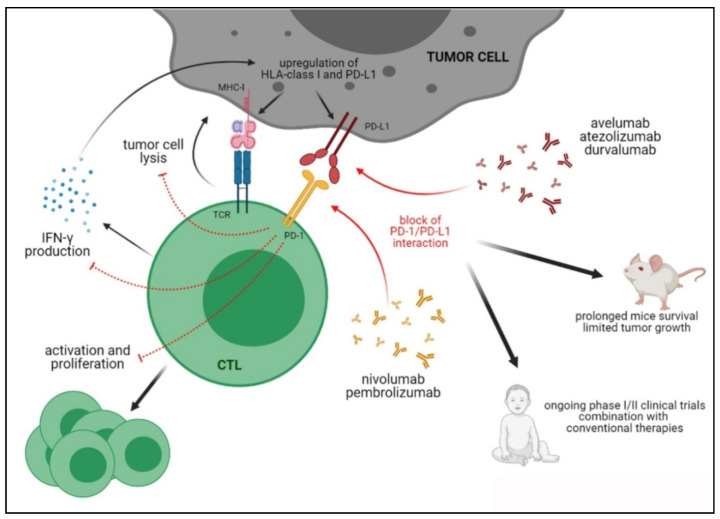
Immune suppression mediated by PD-1/PD-L1 axis and therapeutic activities of anti-PD-1/PD-L1 blocking antibodies in preclinical and clinical studies.

**Figure 2 cells-10-00927-f002:**
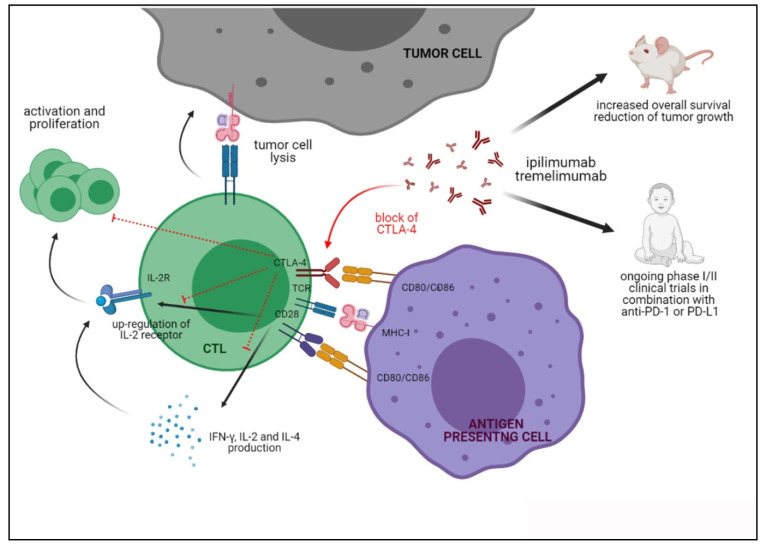
Functions of CTLA-4 upon interaction with CD80/CD86 and the activities of anti-CTLA-4 blocking antibodies in preclinical and clinical studies.

**Figure 3 cells-10-00927-f003:**
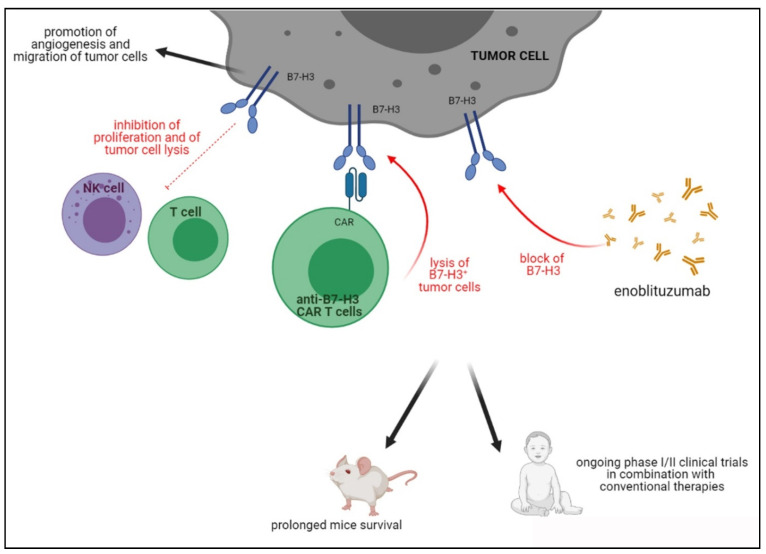
Activities of B7H3 in TME and inhibition driven by blocking antibodies or B7H3 CAR T cells in preclinical and clinical studies.

**Figure 4 cells-10-00927-f004:**
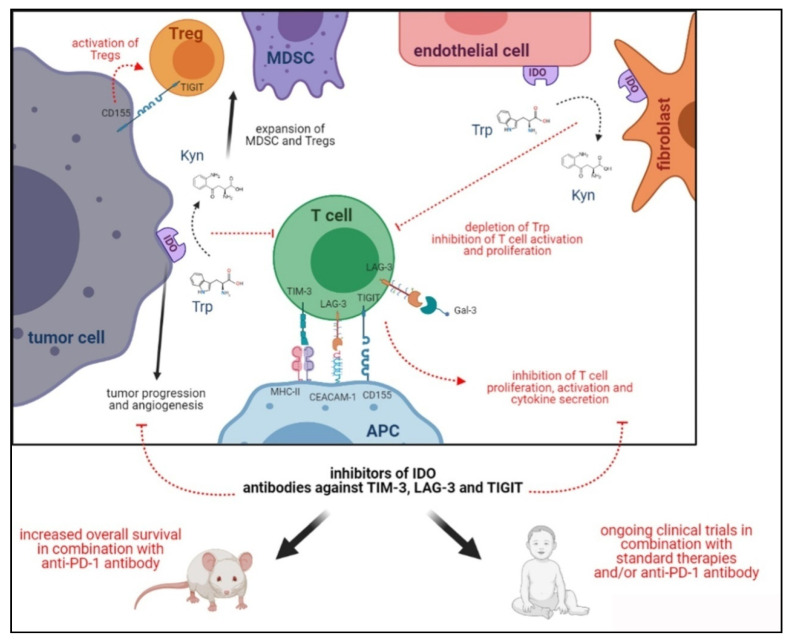
Expression of IDO-1, TIM-3, LAG-3 and TIGIT in TME and their immune suppressive activities.

## Data Availability

Not applicable.

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
