# Peer review of "Immune Checkpoints in Pediatric Solid Tumors: Targetable Pathways for Advanced Therapeutic Purposes"

_cells, 2021, doi:10.3390/cells10040927_

Round 1

Reviewer 1 Report

The review article entitled ''Immune checkpoints in pediatric solid tumors: targetable pathways for advanced therapeutic purposes'' is very interesting with the recent advancements in the field of check inhibitors.

  1. The authors mentioned the importance of TGFbeta in general. With the recent advancements in the field and publications suggesting the possibility of using TGFbeta therapy with checkpoint inhibitors is noteworthy to be mentioned in the review.
  2. With the emerging technology of CRISPR-Cas9, the authors can give a future perspective. 
  3. As the solid tumors microenvironment is very versatile with an influx of various cytokines, TILs etc. Can the use of bispecific antibodies be an alternate approach for treatment. The authors could shed some light in the discussion. 

Author Response

Reviewer #1

The review article entitled ''Immune checkpoints in pediatric solid tumors: targetable pathways for advanced therapeutic purposes'' is very interesting with the recent advancements in the field of check inhibitors.

  1. The authors mentioned the importance of TGFbeta in general. With the recent advancements in the field and publications suggesting the possibility of using TGFbeta therapy with checkpoint inhibitors is noteworthy to be mentioned in the review.

We thank the Reviewer for this suggestion. We added a sentence in the Introduction, paragraph 1 (page 2) and then we addressed possible combination of IC inhibitors with drugs targeting TGF-β in sub-section 4.1.1 (page 9) and paragraph 4.2 (page 11). A comment has been added in the Discussion (page 17-18).

  1. With the emerging technology of CRISPR-Cas9, the authors can give a future perspective.

We agree with this comment, and we address this issue in the Discussion (page 17-18).

  1. As the solid tumors microenvironment is very versatile with an influx of various cytokines, TILs etc. Can the use of bispecific antibodies be an alternate approach for treatment. The authors could shed some light in the discussion.

We thank the Reviewer for this comment. Indeed, we have addressed the role of bi-specific molecules targeting IC and cytokines in sub-section 4.1.1 (page 9) and paragraph 4.2 (page 11). Moreover, a comment has been added in the Discussion (page 17-18).

Reviewer 2 Report

The review is excellent and it's an important contribution in this new field. The scientific content and the figures are very clear and helpful for improving the knowledge about the role of the immune check-points in pediatric cancer. The review represents and excellent update and it contributes to a better comprehension of the pathogenic basis for this new promising therapy, even for clinicians.  Only minor spell changes in few words are proposed in some paragraphs.

Author Response

The review is excellent and it's an important contribution in this new field. The scientific content and the figures are very clear and helpful for improving the knowledge about the role of the immune check-points in pediatric cancer. The review represents and excellent update and it contributes to a better comprehension of the pathogenic basis for this new promising therapy, even for clinicians.  Only minor spell changes in few words are proposed in some paragraphs.

We sincerely thank the Reviewer for the appreciation of our work. We have done the suggested corrections in the text.